# Comparison of Associations between MIND and Mediterranean Diet Scores with Patient-Reported Outcomes in Parkinson’s Disease

**DOI:** 10.3390/nu14235185

**Published:** 2022-12-06

**Authors:** Devon J. Fox, Sarah JaeHwa Park, Laurie K. Mischley

**Affiliations:** 1Parkinson Center for Pragmatic Research, Seattle, WA 98133, USA; 2Bastyr University Research Institute, Bastyr University, Kenmore, WA 98028, USA; 3Translational Bioenergetics Laboratory, Department of Radiology, University of Washington, Seattle, WA 98105, USA

**Keywords:** disease modification, neuroprotection, parkinsonism, non-motor symptoms, comparative effectiveness

## Abstract

The Mediterranean (MEDI) and Mediterranean-DASH Intervention for Neurodegenerative Delay (MIND) diets have been associated with a reduced risk of Parkinson’s disease (PD) diagnosis. However, studies evaluating whether these diets are associated with disease progression in those patients already diagnosed are lacking. The objective of this study was to evaluate whether MIND and MEDI scores were associated with improved patient-reported outcomes. Additionally, we sought to explore which questions on the MIND and MEDI scales were more strongly correlated with PD symptom severity. Data were obtained from the ongoing Modifiable Variables in Parkinsonism study, using patient-reported outcomes in Parkinson’s disease (PRO-PD) as the primary measure for symptom severity, and MIND and MEDI scales for diet score. After adjusting for age, gender, income, and years since diagnosis, for each 1-point increase in the MIND and MEDI scores, PRO-PD scores were 52.9 points lower (95%CI: −66.4, −39.4; *p* < 0.001) and 25.6 points lower (95%CI: −37.2, −14.0; *p* < 0.001), respectively (N = 1205). This study suggests MIND and MEDI scores are associated with fewer patient-reported symptoms over time, with each MIND point being twice as strong as a MEDI point in reducing symptom severity. Future dietary intervention trials should consider the MIND diet as a therapeutic strategy for improving long-term PD outcomes.

## 1. Introduction

Parkinson’s disease (PD) is the second most common neurodegenerative disorder and is an insidious, debilitating disease, characterized by both motor and non-motor disturbances. Diagnosis is based on cardinal motor features such as tremors, bradykinesia, rigidity, and postural instability; however, motor symptoms are now understood to manifest late in the disease [1,2]. Non-motor symptoms (NMS) of fatigue, cognitive changes, mood disorders, sleep disturbances, hyposmia, gastrointestinal manifestations, and autonomic dysfunction are proposed to be prodromal markers of PD [3,4,5,6]. They often long precede the usual motor symptoms [1,2] and persist throughout the disease [7]. Previous population-based cohort studies have reported that 100% of people with PD (PwP) presented with at least one NMS [7,8,9]. Recent studies have focused on NMS for the prediction and diagnosis of PD [3,5,6] because prodromal PD (pPD) is difficult identify and manage without subjective or patient-reported outcome (PRO) measures [1]. In the absence of biomarkers and without objective measures of pain, fatigue, apathy, and other early NMS, subjectively reported outcomes are essential for PD symptom tracking early in the disease. Early disease modification efforts are not likely to result in measurable motor improvements, which are minimal and often well-managed by dopamine replacement strategies. Subjectively reported NMS and PRO may be especially useful outcome measures for early disease-modifying interventions.

There is a robust and growing body of evidence that dietary habits throughout midlife are associated with the risk of developing PD. Traditional epidemiological data, focusing on incidence and probability, generally suggests a diet high in dairy, meat, refined pastries, and fried food is associated with an increased risk of diagnosis [10,11]. Additionally, they indicate that a diet high in fresh fruits and vegetables, whole grains, legumes, nuts and seeds is associated with a reduced risk of diagnosis [11,12,13,14,15]. Before embarking on a clinical trial of diet as a therapeutic strategy, it is essential that researchers are able to define and quantify adherence.

The Mediterranean (MEDI) and Mediterranean-DASH Intervention for Neurodegenerative Delay (MIND) are two popular diets among PD researchers due to their historical use in the treatment of Alzheimer’s disease and established rating scales. In general, both the MEDI and MIND diets encourage a higher consumption rate of olive oil, fresh fruits and vegetables, fish, poultry, beans, nuts, and wine, while discouraging the consumption of foods such as red meat, butter or margarine, and pastries or sweets. Though similar, there are important differences in the MEDI and MIND scales to note. Namely, the MIND diet rewards higher intake of green leafy vegetables, berries, and beans only, while penalizing higher intake of cheese, butter, and margarine. The MEDI diet, conversely, rewards higher consumption of all vegetables, fruit, and legumes without specification for green leafy vegetables, berries, or beans, and minimizes most dairy while allowing cheese. In the MIND diet but not the MEDI diet, whole grains are rewarded, and fast, fried foods are penalized, along with all pastries and sweets. In the MEDI diet but not the MIND diet, dishes seasoned with sofrito seasoning are rewarded, and sweet or carbonated beverages along with only commercial (not homemade) pastries and sweets are penalized. The cut-off point for the amount of consumption required to score one point in each component of the questionnaires also differs between the two diets. For example, to receive one point in the MIND diet, non-fried fish must be consumed ≥1 meal per week and non-fried poultry must be consumed ≥2 meals per week for another point. In comparison, the MEDI diet requires that fish or shellfish must be consumed ≥3 servings per week to receive that same point and answering “yes” to preferential consumption of chicken, turkey, or rabbit meat earns one point without specification for quantity. The MEDI diet does not discriminate against fried versus non-fried meats, and the MIND diet does not reward shellfish or rabbit meat. The amounts consumed for wine and nuts also differ (≥7 glasses and ≥3 servings per week, respectively, for the MEDI diet and 2–7 glasses and ≥5 servings per week, respectively, for the MIND diet). Though some differences are subtle, they still impact the overall diet score in each questionnaire [16,17].

The MEDI diet is thought to be neuroprotective due to its high antioxidant and anti-inflammatory properties [18,19,20]. A large prospective observational population-based study of 49,261 Swedish women demonstrated greater MEDI adherence in middle age, an effect which was associated with a reduced risk of PD diagnosis [18]. For individuals over the age of 65, each one unit increase in the adherence score, using a 10-point scale proposed by Trichopoulou et al. [21], was associated with a 29% risk reduction (HR = 0.71, 95%CI: 0.57–0.89) [18]. Similarly, diet quality and risk of PD diagnosis were evaluated in the prospective population-based Rotterdam Study (N = 9414), which concluded that Mediterranean-type diets appeared to have a protective effect [14]. These findings were consistent with U.S. population studies [14,15], furthering support for an association between higher MEDI diet adherence and reduced risk of PD diagnosis.

The literature on the preventative effects of the MIND diet against Alzheimer’s disease (AD) [22,23,24] and cognitive decline [17] has been promising thus far; however, there has been a paucity of studies investigating the diet’s utility for PD until recently. Several studies have shown a correlation between MEDI [14,25] and MIND [25] diet adherence and the risk of PD. In people aged 59 to 97 years without parkinsonism at baseline, there was a 13% reduced risk of developing parkinsonism per one unit increase in the MIND score. In separate analyses, there was a 3% reduced risk of developing parkinsonism from the MEDI diet [26]. 

Several studies have supported a correlation of slowed progression in motor symptoms with both the MEDI [14] and MIND [25,26] diets. A U.S. case–control study between PwP (N = 257) and control participants (N = 198) found that, not only was higher MEDI diet adherence associated with a reduced risk of PD diagnosis, there was also a correlation between diet score and PD age-at-onset [14]. In this study, MEDI diet adherence was observed to be higher in controls than PwP participants, and a higher MEDI diet score was associated with a later age of PD onset (β = 1.09, *p* = 0.010), with an average age of 61.7 years [14]. The ongoing Rush Memory and Aging Project (MAP) study found a slower rate of parkinsonism progression was more strongly associated with higher adherence to the MIND diet (β = −0.008; SE = 0.0037; *p* = 0.04) than the MEDI diet (β = −0.002; SE = 0.0014; *p* = 0.06) [26]. Similarly, in a Canadian cross-sectional study, a higher MIND diet score was most significantly associated with later age of onset, with differences of up to 17.4 years between the highest and lowest tertiles of diet scores [25].

The Movement Disorder Society (MDS) criteria was utilized to evaluate the association between pPD probability and MEDI diet score in the Hellenic Longitudinal Investigation of Aging and Diet (HELIAD) Study [27], a large, multidisciplinary, population-based Greek study (N = 1731) [28]. In PD-free individuals over the age of 65, the highest quartiles of MEDI diet score were associated with an approximately 21% lower probability score for pPD and less prevalence of the non-motor markers of pPD, depression, constipation, urinary dysfunction, and daytime somnolence [27]. Similarly, a larger study using cohorts from the Nurses’ Health Study (NHS) and the Health Professionals Follow-up Study (HPFS) (N = 47,679) found that a higher score in the alternate MEDI (aMEDI) diet was inversely associated with individual pPD features of depression, constipation, and daytime somnolence [29]. These studies support a correlation between higher adherence to the MEDI diet and lower probability of pPD; no studies were found assessing the relation between MIND diet and pPD probability. 

There has been a limited number of studies investigating the impact of the MIND diet specifically in PD-associated NMS [25,26,30]. To our knowledge, no studies have compared the effects of MEDI and MIND diet adherence on motor and non-motor symptom severity. There are enough differences between these diets that, prior to disease onset, there is a need for modification studies to confirm prospectively if adherence to the MEDI and MIND diets correlates with better PRO over time in population-based studies. This study utilized the Patient-Reported Outcomes in Parkinson’s Disease (PRO-PD), a subjective scale of 33 symptoms as a marker for PD severity [31]. PRO measures reflect the patient’s perspective of disease severity, may be more sensitive early in disease, and are useful in dietary intervention research [32]. The goal of this study was to evaluate whether diet score was associated with improved PRO-PD, and to explore and compare each scale, and determining which questions on the MEDI and MIND scales were more strongly correlated with PD symptom severity in a cohort of PwP.

## 2. Materials and Methods

This is a cross-sectional analysis of participants enrolled in the ongoing Modifiable Variables in Parkinsonism (MVP) Study (Clinical Trial #NCT02194816). The study was conducted according to the guidelines of the Declaration of Helsinki and approved by the Institutional Review Board of Bastyr University (IRB #13A-1332, 1 December 2021). All participants acknowledged reading and agreeing to the study information sheet, and which was drawn up according to study protocol and approved by the IRB. MVP study participants are emailed a link twice a year, each survey taking approximately 60–90 min in total. Participants do not need to complete the survey in one sitting. Study data are collected and managed using Research Electronic Data Capture (REDCap) tools hosted at Bastyr University [33]. REDCap is a secure, web-based application designed to support data capture for research studies, providing: (1) an intuitive interface for validated data entry; (2) audit trails for tracking data manipulation and export procedures; (3) automated export procedures for seamless data downloads to common statistical packages; and (4) procedures for importing data from external sources.

The 15-question MIND and validated 14-question MEDI [16] diet questions were inserted into the diet portion of the larger MVP Study with IRB approval in 2020. Surveys reported between the dates of 1 January 2020 to 1 July 2022 were included for analysis. Because there is no way to confirm the diagnosis of idiopathic PD (iPD) via this survey, and because the primary outcome measure was symptom burden and not diagnosis, individuals with all forms of parkinsonism were included in this analysis. Secondary analysis evaluated whether the results changed if the analysis was restricted to iPD only. Gender was self-reported (NA = 65), whereas age and years with PD were calculated using patients’ their self-reported date of birth and date of diagnosis, as well as the date of the survey. These results were rounded to the nearest year. Participants were excluded who were outside of the range values, i.e., less than 18 years old or greater than 100 years old. Participants with missing dietary surveys (NA = 146) were excluded from analysis. Dopa equivalents were calculated according to the recommendations of Schade et al. (2020) [34]. Exercise was recorded as the number of days the participant engaged in 30 min or more of physical activity. 

The primary outcome measure of PD severity in the MVP study is the PRO-PD, a questionnaire of 33 common PD symptoms, each assigned a slider bar. Participants are asked to move the tab on an unnumbered slider bar in accordance with that symptom’s severity over the one week prior, with the far left representing optimal health (0) and the far right representing debilitation(100). Total PRO-PD score was calculated as the cumulative score of 33 slider bars assessing the common symptoms of PD, all weighted equally, with a maximum score of 3300. The PRO-PD has previously been shown to correlate with the number of years elapsed since PD diagnosis, quality of life and legacy measures of PD, such as Hoehn and Yahr (H-Y), Unified PD Rating Scale (UPDRS), Parkinson’s Disease Questionnaire-39 (PDQ-39), and Timed-Up-&-Go (TUG). However, the non-motor subset of the PRO-PD (PRO-PD(nm)) was highly correlated with the non-motor symptom scale (NMSS) [30].

Non-motor symptoms included constipation, lack of motivation, depression, loss of interest, anxiety, fatigue, daytime sleepiness, temperature dysregulation, orthostatic hypotension, visual disturbances, insomnia, REM sleep behavior disorder, muscle pain, drooling, memory impairment, comprehension disability, hyposmia, sexual dysfunction, urinary dysfunction, and hallucinations [35]. The motor symptoms sub-score included handwriting, slowness of movement, tremors, stooping posture, difficulty of walking, speech, rising from a seating position, dressing, restless legs, falling, balance and freezing (Appendix A). 

MEDI score was calculated by using a validated 14-item questionnaire, [16]. Tomato sauce was included in a positive response for sofrito. A positive response equals 1 point, and a negative response equals 0 points. MIND diet score was calculated using a 15-item subcategory adapted from the original paper that defines dietary components using items in the food frequency questionnaire (FFQ) of the parent cohort study, which assigned scores of 0, 0.5, or 1 point according to the amount consumed in the corresponding time frame [17] (Appendix A). 

### Statistical Analysis

All PRO-PD data was analyzed using Rstudio [36]. Skew and kurtosis were tested for predictor and outcome variables, being determined acceptable if values were between −1 and +1 [37]. Cross-sectional linear regressions were conducted on predictors of MEDI and MIND diet scores with the outcomes of total PRO-PD score, non-motor and motor PRO-PD sub-scores. Adjustments were made for years elapsed since PD diagnosis, age, gender and income. The rate of symptom severity were calculated by self-reported date of diagnosis and PRO-PD score. Those with years with PD values of zero were excluded from this regression (N = 61). Linear regression slopes were compared between MEDI and MIND via a Wald and *t* test to the determine statistically significant difference in slope for total PRO-PD score, sub-scores and progression.

Individual items of each diet were assessed for association with PRO-PD score, and significant associations were used in a restricted analysis for association with PRO-PD score. NMS were assessed for correlation with MEDI and MIND diet scores, with adjustments made for age, gender, income, and years since diagnosis. When both diets were statistically significant for correlation with diet score, a Wald and *t* test was conducted to determine which diet reduced symptom severity more.

## 3. Results

The sample was composed of 1205 participants, with ages ranging from 36 to 90 with an average age of 66.4 years. The cohort was mostly female, Caucasian, had an annual income greater than $60,000 and a higher education degree. The average H-Y score was 1.9, time elapsed since PD diagnosis was 7.2 years, and tge PRO-PD score was 844, with a non-motor PRO-PD score of 422 and a motor PRO-PD score of 301 (Table 1). The average MEDI score was 7.8, and the MIND diet was 9.8 (Appendix A). The correlation between MEDI and MIND scores was statistically significant (β = 0.784, R^2^ = 0.4582, dof = 956, *p* < 0.001). 

### 3.1. PRO-PD Score by Diet Score

When adjusted for age, gender, income and years since diagnosis, total PRO-PD score decreased by 25.6 (37.2–14.0) points per one point increase in MEDI score (R^2^ = 0.1982, dof = 961, *p* < 0.001). The total PRO-PD score decreased by 52.9 (66.4–39.4) points per point increase in MIND diet score (R^2^ = 0.2207, dof = 964, *p* < 0.001). Non-motor PRO-PD sub-score had a decrease of 13.0 (19.1–6.94) per point increase in MEDI score, (R^2^ = 0.1582, dof = 961, *p* < 0.001) and there was a decrease of 27.3 (34.4–20.2) points per point increase in MIND score (R^2^ = 0.1849, dof = 964, *p* < 0.001). Motor PRO-PD sub-score had a decrease of 9.78 (14.3–5.23) points per point increase in MEDI score (R^2^ = 0.2173, dof = 961, *p* < 0.001), and there was a decrease of 19.8 (25.1–14.5) points per point increase in MIND score (R^2^ = 0.2344, dof = 964, *p* < 0.001) (Figure 1). 

### 3.2. PRO-PD Progression and Diet

A total of 61 participants were excluded (years with PD = 0). The yearly increase in PRO-PD score when adjusted for MEDI score, age, gender, and income was 30.7 (25.4–35.9) points (R^2^ = 0.2073, dof = 961, *p* < 0.001). When adjusted for MIND diet score, age, gender, and income, PRO-PD increased by 27.3 (22.2–32.3) points per year with PD (R^2^ = 0.2207, dof = 964, *p* < 0.001) (Figure 1). The difference in slope between the two diet scores was non-significant.

The non-motor PRO-PD score increase per year with PD was 13.2 (10.6–15.7) points (R^2^ = 0.1469, dof = 1091, *p* < 0.001). For MEDI, the non-motor PRO-PD score increased by 13.2 (10.4–15.9) points (R^2^ 0.1677, dof = 961, *p* < 0.001), while for those adhering to the MIND diet, the non-motor PRO-PD score increased by 11.6 (8.94–14.2) points each year with PD (R^2^ = 0.1941, dof = 964, *p* < 0.001).

The motor PRO-PD score increase per year with PD was 13.0 (11.0–14.9) points (R^2^ = 0.2031, dof = 1091, *p* < 0.001). For MEDI, the motor PRO-PD score increased by 13.0 (10.9–15.0) points each year (R^2^ = 0.2261, dof = 961, *p* < 0.001). Those that adhered to the MIND diet had an increase in motor PRO-PD score of 11.7 (9.72–13.7) points per year (R^2^ = 0.243, dof = 964, *p* < 0.001). All regressions were adjusted for age, gender, and income.

### 3.3. PRO-PD Symptoms and Diet

Increased MEDI scores had a significant inverse correlation with NMS of constipation (*p* = 0.0005), motivation (*p* < 0.001), depression (*p* = 0.003), withdrawal (*p* < 0.001), anxiety (*p* = 0.01), fatigue (*p* = 0.04), daytime sleepiness (*p* = 0.001), visual disturbances (*p* = 0.03), insomnia (*p* = 0.002), muscle pain (*p* = 0.003), forgetfulness (*p* < 0.001), comprehension (*p* = 0.002), and sexual dysfunction (*p* = 0.02). MIND score had significant reductions in all NMS assessed (constipation: *p* < 0.001, motivation: *p* < 0.001, depression: *p* < 0.001, withdrawn: *p* < 0.001, anxiety: *p* < 0.001, fatigue: *p* < 0.001, daytime sleepiness: *p* < 0.001, visual disturbance: *p* < 0.001, insomnia: *p* < 0.001, REM sleep behavior disorder: *p* = 0.002, muscle pain: *p* < 0.001, forgetfulness: *p* < 0.001, comprehension: *p* < 0.001, sexual dysfunction: *p* < 0.001, urinary: *p* < 0.001, hallucinations: *p* = 0.02). All regressions were adjusted for age, gender, income, and years since diagnosis (Table 2). 

### 3.4. MIND vs. MEDI Comparison

Regression coefficients of fully adjusted models were compared via a Wald test and paired *t* test. MIND showed a significant reduction in regression slope compared to MEDI in total PRO-PD (F = 14.5, *p* < 0.001), non-motor PRO-PD (F = 13.6, *p* < 0.001) and motor PRO-PD (F = 13.8, *p* < 0.001) scores. When the rates of progression of the total PRO-PD score were compared between MEDI and MIND diets, there was a non-significant difference (F = 2.77, *p* = 0.1, n = 1144). MEDI and MIND diets significantly differed on the NMS severity correlation of constipation (F = 6.96, *p* = 0.01), anxiety (F = 6.18, *p* = 0.02), fatigue (F = 6.58, *p* = 0.01), insomnia (F = 5.51, *p* = 0.02), drool (F = 5.91, *p* = 0.03), forgetfulness (F = 6.92, *p* = 0.01), sexual dysfunction (F= 4.58, *p* = 0.04), and urinary problems (F = 7.54, *p* < 0.001). A linear regression was used to adjusted for age, gender, income and years since diagnosis with a Holm–Bonferroni correction (Table 2).

Individual items from each diet scoring system were assessed for correlation to symptom severity by linear regressions adjusted for age, gender, income and years since diagnosis with a Holm–Bonferroni correction. The MEDI items that had significantly reduced PRO-PD scores were consumption of >2 servings of vegetables a day (*p* = 0.02), less than 1 serving of red meat per day (*p* = 0.04), less than 1 sweet or carbonated beverages a day (*p* < 0.001), and greater than 3 or more servings of nuts per week (*p* = 0.005) (Appendix A). The habits significantly correlated to the MIND diet were as follows: the consumption of more than 6 servings of green leafy vegetables (*p* < 0.001); more than 1 potion a day of other vegetables (*p* < 0.001); more than 2 servings of berries (*p* = 0.03); more than 5 servings of nuts (*p* < 0.001); less than 1 teaspoon a day of butter (*p* < 0.001), less than 1 serving a week of cheese (*p* < 0.001), and less than 4 meals a week with red meat (*p* < 0.001) (Appendix A). The items that yielded non-significant correlations with a higher PRO-PD score were consumption of sofrito (red sauce) for more than 2 servings a week, consumption of non-fried poultry, pastries and sweets, and fast fried food. 

When restricted to only significant items, MEDI had a significant reduction in total PRO-PD (74.7, 95%CI: 102–47.9 *p* < 0.001), non-motor PRO-PD (36.2, 95%CI: 50.2–22.2, *p* < 0.001) and motor PRO-PD (30.3, 95%CI: 40.7–19.8, *p* < 0.001) scores per point increase in diet score. MIND also had significant reduction in PRO-PD (106, 95%CI: 129–84.3, *p* < 0.001), non-motor PRO-PD (54.9, 95%CI: 66.5–43.3, *p* < 0.001), and motor PRO-PD (38.4, 95%CI: 47.1–29.7, *p* < 0.001) scores per point increase in diet score. When regression coefficients were compared, they showed that MIND had a significantly reduced slope compared to MEDI for total PRO-PD (F = 12.4, *p* < 0.001), non-motor PRO-PD (F = 14.2, *p* < 0.001) and motor (F = 7.33, *p* = 0.007) scores.

Additional covariates of exercise and dopamine equivalent were included, and regressions were assessed for differences in significance. MEDI and MIND diet scores remained significantly associated with total PRO-PD (MEDI: *p* = 0.04; MIND: *p* < 0.001), non-motor PRO-PD (MEDI: *p* = 0.04; MIND: *p* < 0.001), and motor PRO-PD (MEDI: *p* = 0.046; MIND: *p* < 0.001). The Wald test results between the regressions with the additional covariates remained significant for total PRO-PD (*p* < 0.001), non-motor PRO-PD (*p* < 0.001), and motor PRO-PD (*p* = 0.001).

## 4. Discussion

The goal of this cross-sectional observational study was to evaluate the association of and statistical difference between MEDI and MIND diet scores in this Parkinson cohort, with symptom severity as measured by patient-reported outcomes. After adjusting for age, gender, income, and years with PD, diet scores were significantly inversely correlated with total, non-motor and motor PRO-PD scores. Adherence to the MIND diet produced significantly reduced total, non-motor, and motor PRO-PD scores compared to MEDI, suggesting the MIND diet has a great impact on symptom severity. All NMS had significant reductions with higher MIND diet scores, while MEDI had significant reductions for most NMS, with MIND reducing symptom severity to a significantly greater extent than MEDI. MIND diet scores produced significantly reduced symptom severity in constipation, anxiety, fatigue, insomnia, forgetfulness, and sexual dysfunction compared to MEDI. MEDI and MIND diets had similar rates of progression. Progression was estimated for this cross-sectional study using current symptoms and date of diagnosis; longitudinal data collection is underway and will, in time, describe whether changes in behavior result in a change in slope. 

Exploratory analysis was conducted on the individual diet questions with the aim of elucidating specific dietary modifications that contribute to the significant difference in symptom severity between the two scales, and guiding future dietary research for the development of dietary interventions in neurodegenerative diseases. This analysis showed that specific diet categories significantly reduce symptom severity. Higher consumption levels of butter/margarine, red meat, cheese, fast food, carbonated/sweet beverages, and pastries and sweets were associated with greater severity of PD symptoms. Conversely, greater consumption of nuts, vegetables, berries, beans and non-fried fish were associated with lesser severity of PD symptoms. Most studies to date have focused on each diet separately; this is the one of the first studies to look, by question, at which components of these diets were most strongly correlated with PD outcomes over time and subsequently compare diets. 

To our knowledge, this is the first study that compares the effects of the MIND and MEDI diets on both motor and non-motor symptom severity in PD. Previous studies have demonstrated that both diets reduce PD risk [13,14,15,18,25,26] or delay the onset of motor symptoms [14,25,26]. In previous studies, the MEDI diet has been associated with improved NMS of constipation and indigestion [38], cognitive impairment [39], and depression [40,41,42] in PwPD. A higher MIND diet score was substantially associated with slower cognitive decline with aging in the MAP Study [17], and improved verbal memory scores in a population-based study of American women [43]. However, results still vary in this population, potentially due to differences in assessment tools and statistical power [30]. In this study, the non-motor PRO-PD sub-scale used assessed “memory/forgetfulness” and “comprehension” as cognitive measures, which were both found to be significantly improved from the MEDI diet and, more so, the MIND diet. These data were consistent with findings of recent studies that suggested the MIND diet was more strongly correlated with reduced risk and slower progression of parkinsonism compared to the MEDI diet [18,26]. Interestingly, sex-specific effects between the MIND and MEDI diets have also previously been reported. Metcalfe-Roach et al. (2021) [24] found that the MIND diet was most significantly associated with later age-at-onset in the female subgroup, more than three times greater than that of the male or MEDI diet subgroups, suggesting that the dietary components in the MIND diet (e.g., leafy green vegetables and berries) may potentially be more beneficial for delaying the onset of motor symptoms in female populations [24].

The biochemical effect of nutrition on neurodegenerative disease has been recently reviewed in a systematic review, which reported that Mediterranean-like diets may protect neurons and slow neurodegeneration via several mechanisms of action [17]. Namely, the increased intake of polyphenols [44], resveratrol [45], omega-3 fatty acids [46], vitamin E, vitamin C [47], and carotenoids [47,48], from fruits and vegetables, olive oil, and red wine, has beneficial antioxidant and anti-inflammatory properties [19]. Another potential mechanism of action is via the modulation of gut microbiota by a diet high in fiber and polyphenols, which both increase the amount of beneficial bacteria in the microbial composition of the gut [49,50]. Although both diets may be beneficial in slowing the neurodegenerative processes, the vast number of dietary components in these diets, with several overlapping food groups, makes it difficult to understand which food groups, exactly, may be responsible for the effects observed in this study. The main differences between these diets, specifically the emphasis on leafy green vegetables and berries in the MIND diet, as well as the exclusion of dairy products including cheese, butter, and margarine, should serve as a point of focus in future studies.

The cross-sectional nature of this method does not allow causal inference due to the lack of temporal analysis. The nature of this method does not allow for monitoring changes in dietary habits over time. Future research should utilize prospective studies to assess the impact of dietary habits on PD symptom severity. 

The sample consisted largely of adults that adhered to an at least moderate MEDI or MIND diet. Of the 1205 participants, only 16 scored in the bottom third of diets. This study was also largely white, wealthy, U.S.-based, and 59% female. Gender was not an effect modifier in this study. A limitation inherent to all non-randomized studies is that the choice and ability to adhere to a MIND or MEDI diet may be attributable to other co-occurring protective variables, such as gym memberships, or better pharmaceutical or provider access. Furthermore, limitations exist in this study, including the lack of validation for the MIND diet scale and the PRO-PD scale. The dietary scales used in this study are brief questionnaires and are not part of a larger FFQ. A validation study should be conducted to ensure the accuracy of this scale. The MEDI diet contains a question regarding sofrito and, due to the characteristics of majority of this population and the self-reported nature of this study, tomato sauce, a non-traditional version of sofrito, is included in a positive response. 

## 5. Conclusions

This observational study suggests that greater MIND and MEDI diet scores were associated with fewer patient-reported PD symptoms, with each MIND point being twice as strong as an MEDI point. If these data are reproducible in more diverse populations, dietary intervention studies should focus on increasing MIND scores as a therapeutic strategy. Regardless of the label applied, there was an agreement between both scales that the best PD outcomes were found in those who consumed fresh fruit and vegetables, nuts, beans, non-fried fish and avoided butter, margarine, cheese, fast fried food, pastries and sweets, red meat, pork, and soda. Until a PD-specific diet is described, and an adherence scale developed and validated, these data suggest adherence to the MIND diet, more than MEDI, is associated with the accumulation of fewer patient-reported symptoms over time. 

## Figures and Tables

**Figure 1 nutrients-14-05185-f001:**
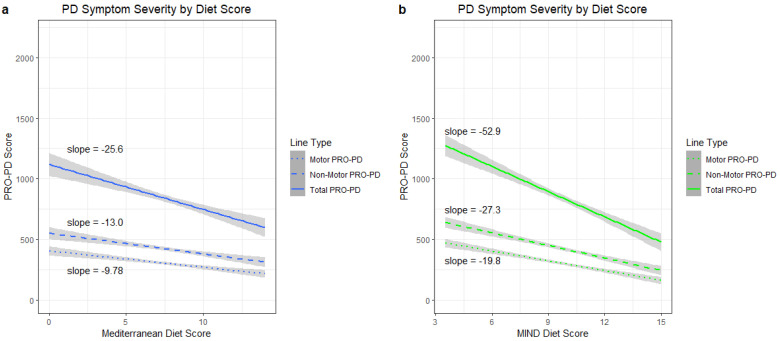
(**a**) Linear regression of PRO-PD score by Mediterranean Diet Score adjusted for age, gender, income, and years since diagnosis with non-motor and motor PRO-PD sub-scores. (**b**) Linear regression of PRO-PD score by MIND Score adjusted for age, gender, income, and years since diagnosis with non-motor and motor PRO-PD sub-scores.

**Table 1 nutrients-14-05185-t001:** Sample characteristics including demographics, Parkinsonism severity, and type of parkinsonism.

Characteristics of Study Participants	
	N = 1205
Age, mean (years, SD)	66.4 (8.76)
Years Since Diagnosis (years, SD)	7.19 (5.44)
PRO-PD, mean (year, SD)	844.5 (497.4)
Estimate Hoehn & Yahr	
1-sided symptoms only, minimal disability	533 (45.8%)
Both sides affected, balance is stable	247 (21.2%)
Mild to moderate disability, balance affected	326 (28.0%)
Severe disability, able to walk and stand without help	42 (3.6%)
Confinement to bed or wheelchair unless aided	4 (0.34%
Don’t know	11 (0.95%)
Gender	
Male	474 (39%)
Female	714 (59%)
Non-Binary	3 (0.25%)
NA	14 (1.2%)
Race	
Caucasian	1104 (91%)
Black	8 (0.66%)
Hispanic	22 (1.8%)
Native American	3 (0.25%)
Asian/Pacific Islander	22 (1.8%)
Other	33 (2.7%)
NA	13 (1.1%)
Income	
<$20,000	59 (4.9%)
$20–40,000	107 (8.9%)
$40–60,000	145 (12%)
$60–80,000	188 (16%)
$80–100,000	155 (13%)
$100–150,000	208(17%)
>$150,000	252 (21%)
NA	91 (7.6%)
Education	
Grades 9–11	8 (0.67%)
Completed High School/GED	8.6 (104%)
Technical school certification	5.14 (62%)
Associate Degree	7.5 (90%)
Bachelor’s degree	27 (32%)
Graduate/Professional degree	50 (604%)
NA	0.91 (11%)
Parkinsonism	
Idiopathic PD	1145 (95%)
Parkinsonism	42 (3.5%)
No Dx but increased risk	2 (0.17%)
Other	15 (1.3%)

**Table 2 nutrients-14-05185-t002:** Non-motor symptom correlation with MEDI and MIND diet scores. Correlations were assessed via linear regressions adjusted for age, gender, income, and years since diagnosis with a Holm–Bonferroni post hoc adjustment.

	MEDI		MIND			
Non-Motor Symptoms	β	*p*	β	*p*	F-Statistic	*p*
constipation	−1.08 (−1.68–−0.467)	0.001	−2.11 (−2.83–−1.39)	<0.001	6.96	0.01
motivation	−1.39 (−1.97–−0.809)	<0.001	−2.46 (−3.13–−1.79)	<0.001	3.9	0.06
depression	−0.727 (−1.21–−00.244)	0.01	−1.41 (−1.98–−0.838)	<0.001	2.96	0.1
withdrawn	−1.15 (−1.7–−0.578)	0.001	−1.94 (−2.60–−1.28)	<0.001	2.31	0.1
anxiety	−0.739 (−1.34–−0.134)	0.1	−1.59 (−2.30–−0.890)	<0.001	6.18	0.02
fatigue	−0.675 (−1.31–−0.0348)	0.4	−1.83 (−2.58–−1.07)	0.4	6.58	0.01
daytime sleepiness	−1.04 (−1.66–−0.426)	0.01	−1.91 (−2.64–−1.18)	<0.001	3.86	0.06
visual disturbance	−0.583 (−1.12–−0.0480)	0.4	−1.26 (−1.88–−0.645)	<0.001	2.21	0.2
insomnia	−1.07 (−1.75–−0.399)	0.02	−2.14 (−2.93–−1.35)	<0.001	5.51	0.02
REM sleep behavior disorder	−0.514 (−1.17–0.142)	0.9	−1.20 (−1.98–−0.424)	0.03	1.71	0.2
muscle pain	−0.975 (−1.61–−0.341)	0.03	−1.80 (−2.55–−1.05)	<0.001	3.58	0.06
drool	−0.182 (−0.785−0.422)	0.9	−1.20 (−1.91–−0.498)	0.009	5.91	0.03
memory/forgetfulness	−1.17 (−1.72–−0.623)	0.004	−2.12 (−2.77–−1.48)	<0.001	6.92	0.01
comprehension	−0.801 (−1.31–−0.292)	0.02	−1.34 (−1.93–−0.748)	<0.001	2.46	0.13
sexual dysfunction	−0.906 (−1.69–−0.121)	0.3	−2.16 (−3.07–−1.24)	<0.001	4.58	0.04
urinary symptoms	−0.133 (−0.806–0.539)	0.9	−1.44 (−2.23–−0.649)	0.004	7.54	<0.001
hallucinations	−0.316 (−0.694–0.0619)	0.9	−0.523 (−0.969–−0.0772)	0.2	1.98	0.1

## Data Availability

Data can be requested via https://redcap.bastyr.edu/redcap/surveys/?s=MTND8FM4XY (accessed on 2 December 2022).

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
