# Peer review of "Comparison of Associations between MIND and Mediterranean Diet Scores with Patient-Reported Outcomes in Parkinson’s Disease"

_nutrients, 2022, doi:10.3390/nu14235185_

Round 1

Reviewer 1 Report

The authors present a cross-sectional analysis of the association of the two dietary patterns i.e., MIND and Mediterranean diet, both of which are widely studied for neurodegenerative disorders, with PD-related outcomes and symptoms among participants with PD from an ongoing trail, MVP study. Authors do address an important question on how diet is associated with PD symptoms among patients with parkinsonism and iPD using a diet screener/assessment tool and a PRO-PD measure. However, there are a few suggestions and comments on this presented work.

1.       Title: “Comparison of MIND and Mediterranean Diet Adherence on Patient-Reported Outcomes in Parkinson’s Disease”

Comment: Please do not call diet assessed at a one-time point using a dietary screener/ diet scale questionnaire rather than a validated food frequency questionnaire “diet adherence”. Also, change the title such that it indicated the association/relation more than the comparison. E.g. MIND and Mediterranean Diet association with Patient-Reported Outcomes in Parkinson’s Disease.

2.       Abstract:

Comment:  Please indicate the sample size.

The Medi assessment tool used for this study is validated but the MIND diet one is not. So please edit that. The mind diet questionnaire used in the study was adapted from the first paper published on MIND diet and cognition. Thus, authors should indicate that and call the MIND diet screener/scale/assessment tool something “XXX” and use the same terminology throughout the paper.

3.       Introduction

Line 59-60: “Namely, the MIND diet rewards higher intake of green leafy vegetables and berries, while penalizing all dairy products”

Comment: MIND penalizes only cheese, which is too full-fat cheese, although not clearly mentioned in the original 2015 paper, there is no negative point on milk or yogurt, etc.

Another important to notice is the cut-off points between MIND and Medi, for e.g. for MIND, a point is given when fish/seafood non fired more than 1, however, for Medi the point is given for fish/seafood if more than equal to 1. And this is evident in your diet score distribution also. For the MIND due to different cut-offs, people get points even with low intake thus distribution is narrow compared to MEDI. These cut-offs should be also discussed.

Line 80-81: “there was a 13- 80 42% reduced risk of developing parkinsonism per point increase in MIND score; whereas, higher MEDI diet adherence was associated with a 3% reduced risk [25].”

Comment: As reported in the paper (ref#25), with one unit change in the MIDN diet score there was 13% reduced risk for parkinsonism. 42% is when comparing tertiles (highest vs. lowest). Please edit. Also, one cannot compare the MIND and MEDI results directly as the score range varies, for MIND score is 0-15 and for MEDI it is 0-55. In ref. 25, the authors compared standardized betas later. Edit as appropriate.

Line 120-121: “were more strongly predictive of PD symptom severity in a cohort of PwP.”

Comment: you are looking at association rather than which question is predicting PD symptoms. Please edit.

4.       Methods:

Line 137-138: “The validated MEDI and MIND diet questions were inserted into the diet portion of the 138 larger MVP Study with IRB approval in 2020”

Comment: The MEDI assessment tool has been validated however, the MIND diet questions used in the study are based on the original paper, which used a 144-item questionnaire, so edit that and preferably call this a tool adapted from the original paper as previously mentioned in comment 2.

Line 170-171: “MIND diet 169 adherence was calculated using a 15-item subcategory of the Food Frequency Questionnaire (FFQ), which assigned scores of 0, 0.5, or 1 point according to the amount consumed in the corresponding time frame [23].”

Comment: Again, mention this scoring was in the original paper using FFQ, and this study adapted the scoring matrix. As far as reading the methods, this study does not use FFQ, correct?

Line 180: “Those with years with PD values of zero were excluded from this regression”

Comment: indicate how many.

5.       Statistical procedures can be presented with a subheading. Also, additionally, control for PD medication use and physical activity if available. If not available that should be discussed in the limitations.

6.       If the comparison is one of the aims of this paper, please specify the rationale for the comparison when the diets are so similar.

7.       Results:

In table 1: where presenting n (%), I think for race, income, education, and parkinsonism the numbers are switched.

Also, state the correlation between MIDN and MEDI scores in the analytical sample.

Not sure what is the significance of Figure 1 in the main paper, given the scores are so similar, and the distribution varies mostly because of cut-off points. Maybe move this to supplementary material.

In table 2, add a footnote on the model used and the covariates in the model.

For table 3, multiple comparisons should be addressed. Also, when investigating one question, controlling for others is also important, as eating some specific food groups is correlated. Also, add a column on the number of people who obtained “1” in that question. Else, use the mean score for that question for e.g. in the MIND diet questionnaire.

This table can also be a supplementary file.

8.       Please modify the language around “adherence” throughout out the manuscript.

9.       Discussion:

Line 288: delete “=”

Line 300: “or delay the onset of motor symptoms [15,24,25].”

Comment: ref 25 is more about diet associated with reduced risk of parkinsonism and slower progression of parkinsonian signs. Rewrite this line.

Line 315-6:  “Namely, the increased intake of polyphenols from fruits and vegetables, olive oil, and red wine, which have beneficial antioxidant and anti-inflammatory properties.”

Comment: needs a reference.

Also add references to include other antioxidant nutrients and PD/parkinsonian signs e.g. its not only polyphenols but also carotenoids, vitamin E, and vitamin C in these foods.

Line 324-5:  “The MIND diet, as well as the exclusion of all dairy products, should serve as a point of focus in future studies.”

The exclusion of dairy products is not part of the MIND diet recommendation, so add a reference to support your comment on restricting dairy.

The line “Interestingly, 333 sex-specific effects between the MIND and MEDI diets have previously been reported. Metcalfe-Roach et al (2021) [24] found that the MIND diet was most significantly associated with later age-at-onset in the female subgroup, more than three times greater than that of the male or MEDI diet subgroups, suggesting that the dietary components in the MIND diet (e.g., leafy green vegetables and berries) may potentially be more beneficial for delaying the onset of motor symptoms in female populations [24].”

This piece reads oddly here, maybe move this above in the discussion section and the second last paragraph can be limitations, etc.

In limitations, authors should bring up that the dietary assessment tool is a brief screener rather than the full FFQ and one of the questionnaires has not been published as valid but just adapted from the original paper.   

10.   Conclusion: “Regardless of the label applied, there was universal agreement between the scales that the best PD outcomes were found in those who consumed fresh fruit and vegetables, nuts, beans, non-fried fish and avoided butter, margarine, cheese, fast fried food, pastries and sweets, red meat, pork, and soda. While MIND and MEDI are conveniently established, both included components not associated with better PD outcomes, suggesting neither is optimized for reducing PD symptoms over time.”

Comment: In the first statement, by “universal” what does that mean? as also shown in other studies? If yes, include other references.

Unless considering multiple comparisons, this can be over-concluding.

The second sentence is more of an overstatement with just a cross-sectional association of each very broadly asked question on components. As it is unknown if there are within-food group interactions i.e., the combination of foods making a difference and/or nutrient-genomic/metabolomic aspect helping these other questions come as significant in the model. I would suggest removing this.

Author Response

Reviewer 1:

The authors present a cross-sectional analysis of the association of the two dietary patterns i.e., MIND and Mediterranean diet, both of which are widely studied for neurodegenerative disorders, with PD-related outcomes and symptoms among participants with PD from an ongoing trail, MVP study. Authors do address an important question on how diet is associated with PD symptoms among patients with parkinsonism and iPD using a diet screener/assessment tool and a PRO-PD measure. However, there are a few suggestions and comments on this presented work.

  1. Title: “Comparison of MIND and Mediterranean Diet Adherence on Patient-Reported Outcomes in Parkinson’s Disease”

Comment: Please do not call diet assessed at a one-time point using a dietary screener/ diet scale questionnaire rather than a validated food frequency questionnaire “diet adherence”. Also, change the title such that it indicated the association/relation more than the comparison. E.g. MIND and Mediterranean Diet association with Patient-Reported Outcomes in Parkinson’s Disease.

Thank you for highlighting the appropriateness of verbiage in the diet scales and the cross-sectional nature of this study.  We have adjusted adherence to score where appropriate throughout the manuscript.

Thank you for bringing the comparison and association testing clarification. A main aim of this study was to compare the slopes and intercepts of the outcome measure of patient reported outcomes between the two diet scales to see if there was a significant difference between the two scales.  We have updated language and clarified in the introduction and methods section for this aim.

  1. Abstract:

Comment:  Please indicate the sample size.

Thank you for the suggestion. The abstract has been updated accordingly.

The Medi assessment tool used for this study is validated but the MIND diet one is not. So please edit that. The mind diet questionnaire used in the study was adapted from the first paper published on MIND diet and cognition. Thus, authors should indicate that and call the MIND diet screener/scale/assessment tool something “XXX” and use the same terminology throughout the paper.

Thank you for bringing the lack of validation for the MIND diet scale to our attention. The language has been adjusted throughout the manuscript, including the title, and abstract and an additional comment was made in the limitations.

  1. Introduction

Line 59-60: “Namely, the MIND diet rewards higher intake of green leafy vegetables and berries, while penalizing all dairy products”

Comment: MIND penalizes only cheese, which is too full-fat cheese, although not clearly mentioned in the original 2015 paper, there is no negative point on milk or yogurt, etc.

Another important to notice is the cut-off points between MIND and Medi, for e.g. for MIND, a point is given when fish/seafood non fired more than 1, however, for Medi the point is given for fish/seafood if more than equal to 1. And this is evident in your diet score distribution also. For the MIND due to different cut-offs, people get points even with low intake thus distribution is narrow compared to MEDI. These cut-offs should be also discussed.

Thank you for bringing this to our attention. This sentence has been revised to indicate that the MIND diet penalizes only cheese, butter, and margarine instead of all dairy products. Cut-offs for fish and poultry and other important differences were added.

Line 80-81: “there was a 13- 80 42% reduced risk of developing parkinsonism per point increase in MIND score; whereas, higher MEDI diet adherence was associated with a 3% reduced risk [25].”

Comment: As reported in the paper (ref#25), with one unit change in the MIDN diet score there was 13% reduced risk for parkinsonism. 42% is when comparing tertiles (highest vs. lowest). Please edit. Also, one cannot compare the MIND and MEDI results directly as the score range varies, for MIND score is 0-15 and for MEDI it is 0-55. In ref. 25, the authors compared standardized betas later. Edit as appropriate.

Thank you for bringing this to our attention. We have revised this sentence to no longer read as a direct comparison between the MIND and MEDI diets.

Line 120-121: “were more strongly predictive of PD symptom severity in a cohort of PwP.”

Comment: you are looking at association rather than which question is predicting PD symptoms. Please edit.

Thank you for bringing attention to this interpretation of these articles and the aim.

The language discussing the aim of this paper has been adjusted to bring attention to the comparison between the two scales on PRO-PD and exploratory analysis of individual items of the scales.

  1. Methods:

Line 137-138: “The validated MEDI and MIND diet questions were inserted into the diet portion of the 138 larger MVP Study with IRB approval in 2020”

Comment: The MEDI assessment tool has been validated however, the MIND diet questions used in the study are based on the original paper, which used a 144-item questionnaire, so edit that and preferably call this a tool adapted from the original paper as previously mentioned in comment 2.

Line 170-171: “MIND diet 169 adherence was calculated using a 15-item subcategory of the Food Frequency Questionnaire (FFQ), which assigned scores of 0, 0.5, or 1 point according to the amount consumed in the corresponding time frame [23].”

Comment: Again, mention this scoring was in the original paper using FFQ, and this study adapted the scoring matrix. As far as reading the methods, this study does not use FFQ, correct?

Thank you for these suggestions.  Language has been adjusted to reflect the origins of the MIND diet scale and clarify the development and validity of the scale in the methods and limitations sections.

Line 180: “Those with years with PD values of zero were excluded from this regression”

Comment: indicate how many.

Thank you for bringing attention to this. There was an exclusion of 61 participants with this regression.  It has been noted in the manuscript in the appropriate results section. 

  1. Statistical procedures can be presented with a subheading. Also, additionally, control for PD medication use and physical activity if available. If not available that should be discussed in the limitations.

Thank you for this suggestion. We aimed to be consistent with other papers from this journal in this submission.  We have adjusted the method section to add the subsection of statistical analysis. For additional covariates, we initially aimed to limit the risk of over correction and have explored the suggested variables per your suggestion. We have conducted additional analysis per your suggestions and added description of the results of the added covariates in the manuscript.

  1. If the comparison is one of the aims of this paper, please specify the rationale for the comparison when the diets are so similar.

Thank you for this comment.  Language has been added to the discussion to clarify the aim in this exploratory analysis.

  1. Results:

In table 1: where presenting n (%), I think for race, income, education, and parkinsonism the numbers are switched.

Also, state the correlation between MIDN and MEDI scores in the analytical sample.

Thank you for this correction.  The table has been corrected and the correlation between MIND and MEDI scores added to the results section.

Not sure what is the significance of Figure 1 in the main paper, given the scores are so similar, and the distribution varies mostly because of cut-off points. Maybe move this to supplementary material.

Thank you for the suggestion.  We have moved this table to the supplementary material.

In table 2, add a footnote on the model used and the covariates in the model.

For table 3, multiple comparisons should be addressed. Also, when investigating one question, controlling for others is also important, as eating some specific food groups is correlated. Also, add a column on the number of people who obtained “1” in that question. Else, use the mean score for that question for e.g. in the MIND diet questionnaire.

This table can also be a supplementary file.

 This table has been moved to supplementary material as suggested.  Mean scores have been added to the table.

  1. Please modify the language around “adherence” throughout out the manuscript.

Thank you.  The manuscript has been adjusted to reflect this language

  1. Discussion:

Line 288: delete “=”

This typo has been corrected.

Line 300: “or delay the onset of motor symptoms [15,24,25].”

Comment: ref 25 is more about diet associated with reduced risk of parkinsonism and slower progression of parkinsonian signs. Rewrite this line.

Thank you for your suggestion. This sentence has been rewritten as “These data were consistent with findings of recent tidies that suggested the MIND diet was more strongly correlated with reduced risk and slower progression of parkinsonism compared to the MEDI diet.”

Line 315-6:  “Namely, the increased intake of polyphenols from fruits and vegetables, olive oil, and red wine, which have beneficial antioxidant and anti-inflammatory properties.”

Comment: needs a reference.

Also add references to include other antioxidant nutrients and PD/parkinsonian signs e.g. its not only polyphenols but also carotenoids, vitamin E, and vitamin C in these foods.

Thank you for your suggestion. We have added other nutrients in addition to polyphenols, including resveratrol, omega-3 fatty acids, vitamin E, vitamin C, and carotenoids, with appropriate references.

Line 324-5:  “The MIND diet, as well as the exclusion of all dairy products, should serve as a point of focus in future studies.”

The exclusion of dairy products is not part of the MIND diet recommendation, so add a reference to support your comment on restricting dairy.

Thank you for bringing this to our attention. We have revised this sentence to “…in the MIND diet, as well as the exclusion of dairy products including cheese, butter, and margarine, should serve as a point of focus in future studies.” We agree that the MIND diet does not exclude all dairy products, so we included those penalized in the MIND diet only.

The line “Interestingly, 333 sex-specific effects between the MIND and MEDI diets have previously been reported. Metcalfe-Roach et al (2021) [24] found that the MIND diet was most significantly associated with later age-at-onset in the female subgroup, more than three times greater than that of the male or MEDI diet subgroups, suggesting that the dietary components in the MIND diet (e.g., leafy green vegetables and berries) may potentially be more beneficial for delaying the onset of motor symptoms in female populations [24].”

This piece reads oddly here, maybe move this above in the discussion section and the second last paragraph can be limitations, etc.

 Thank you for your suggestion. This has been moved to lines 350-356.

In limitations, authors should bring up that the dietary assessment tool is a brief screener rather than the full FFQ and one of the questionnaires has not been published as valid but just adapted from the original paper.   

Thank you for pointing these limitations out.  We have added language to highlight these limitations in the study.

  1. Conclusion: “Regardless of the label applied, there was universal agreement between the scales that the best PD outcomes were found in those who consumed fresh fruit and vegetables, nuts, beans, non-fried fish and avoided butter, margarine, cheese, fast fried food, pastries and sweets, red meat, pork, and soda. While MIND and MEDI are conveniently established, both included components not associated with better PD outcomes, suggesting neither is optimized for reducing PD symptoms over time.”

Comment: In the first statement, by “universal” what does that mean? as also shown in other studies? If yes, include other references.

Unless considering multiple comparisons, this can be over-concluding.

The second sentence is more of an overstatement with just a cross-sectional association of each very broadly asked question on components. As it is unknown if there are within-food group interactions i.e., the combination of foods making a difference and/or nutrient-genomic/metabolomic aspect helping these other questions come as significant in the model. I would suggest removing this.

Thank you for bringing this to our attention.  We aim to only conclude what is justified.  Language has been modified to state our conclusions more appropriately. The second sentence has been removed.

Reviewer 2 Report

This article is very short yet clear. The writing is very easy to follow and statistics are well performed. Overall this is a very nice article that focuses an important nutritional aspect related to a popular human neurodegenerative disease. 

However, I do not understand why Tables and Figures have so poor quality. In particularly all tables have various format and font and some are even with very low resolution. It appears they are images copied from somewhere. The figures have issues as well. I suggest to revise all of the tables and figures to make the resolution, font, and format in consistent.

Author Response

This article is very short yet clear. The writing is very easy to follow and statistics are well performed. Overall this is a very nice article that focuses an important nutritional aspect related to a popular human neurodegenerative disease. 

However, I do not understand why Tables and Figures have so poor quality. In particularly all tables have various format and font and some are even with very low resolution. It appears they are images copied from somewhere. The figures have issues as well. I suggest to revise all of the tables and figures to make the resolution, font, and format in consistent.

Thank you for your feedback.  We have corrected the quality and revised the manuscript according to your comment. All figures have been reformatted and edited for consistency and accuracy.  We found several errors that are corrected in the second submission of this manuscript. 

Round 2

Reviewer 1 Report

Most of the comments have been addressed and the edits are thorough. However, there are some minor suggestions or edits needed:

1.       Title suggestion:

“Comparison of the association between MIND and Mediterranean Diet Scores with patient-reported outcomes in Parkinson’s Disease”

2.       Line 191- 194: “adapted from the original of the Food Frequency Questionnaire (FFQ),”

This statement is still not clear: maybe modify as – adapted from the original paper that defines dietary components using food items in the food frequency questionnaire of the parent cohort study……..

3.       You do include sofrito in the score, but given your study population is US based, is it really consumed in this population? In methods, specify if consuming tomato sauce was counted as sofrito in this study population.

4.       Authors still need to consider multiple comparison testing at least for the food group/ component analysis with so many NMS outcomes.

5.       Add a reference at the end of line 355 (Ref is #26 I think).

Reviewer 2 Report

Concerns are all addressed. 

Author Response

Thank you for your time and suggestions for our manuscript.  We appreciate the feedback.